# Bone Marrow Macrophages Induce Inflammation by Efferocytosis of Apoptotic Prostate Cancer Cells via HIF-1α Stabilization

**DOI:** 10.3390/cells11233712

**Published:** 2022-11-22

**Authors:** Veronica Mendoza-Reinoso, Patricia M. Schnepp, Dah Youn Baek, John R. Rubin, Ernestina Schipani, Evan T. Keller, Laurie K. McCauley, Hernan Roca

**Affiliations:** 1Department of Periodontics and Oral Medicine, University of Michigan School of Dentistry, Ann Arbor, MI 48109, USA; 2Department of Urology, Medical School, University of Michigan, Ann Arbor, MI 48109, USA; 3Department of Orthopaedic Surgery, School of Medicine, University of Pennsylvania, Philadelphia, PA 19104, USA; 4Department of Pathology, Medical School, University of Michigan, Ann Arbor, MI 48109, USA

**Keywords:** hypoxia-inducible factor, efferocytosis, bone marrow macrophages, inflammation

## Abstract

The clearance of apoptotic cancer cells by macrophages, known as efferocytosis, fuels the bone-metastatic growth of prostate cancer cells via pro-inflammatory and immunosuppressive processes. However, the exact molecular mechanisms remain unclear. In this study, single-cell transcriptomics of bone marrow (BM) macrophages undergoing efferocytosis of apoptotic prostate cancer cells revealed a significant enrichment in their cellular response to hypoxia. Here, we show that BM macrophage efferocytosis increased hypoxia inducible factor-1alpha (HIF-1α) and STAT3 phosphorylation (p-STAT3 at Tyr705) under normoxic conditions, while inhibitors of p-STAT3 reduced HIF-1α. Efferocytosis promoted HIF-1α stabilization, reduced its ubiquitination, and induced HIF-1α and p-STAT3 nuclear translocation. HIF-1α stabilization in efferocytic BM macrophages resulted in enhanced expression of pro-inflammatory cytokine MIF, whereas BM macrophages with inactive HIF-1α reduced MIF expression upon efferocytosis. Stabilization of HIF-1α using the HIF-prolyl-hydroxylase inhibitor, Roxadustat, enhanced MIF expression in BM macrophages. Furthermore, BM macrophages treated with recombinant MIF protein activated NF-κB (p65) signaling and increased the expression of pro-inflammatory cytokines. Altogether, these findings suggest that the clearance of apoptotic cancer cells by BM macrophages triggers p-STAT3/HIF-1α/MIF signaling to promote further inflammation in the bone tumor microenvironment where a significant number of apoptotic cancer cells are present.

## 1. Introduction

The process of clearing apoptotic cancer cells by macrophages, known as efferocytosis, commonly occurs during tumor progression and fuels the bone-metastatic growth of cancer cells via subsequent pro-inflammatory and immunosuppressive activity [1,2]. Our previous published work reported that bone marrow (BM) macrophage-dependent efferocytosis of apoptotic prostate cancer cells supported skeletal tumor growth through the secretion of pro-inflammatory cytokines, resulting in an immunosuppressive response [3,4]. Recently, a single-cell RNA sequencing study reported that peritoneal macrophage efferocytosis of apoptotic T cells displayed heterogeneous transcriptional activity, including genes associated with a predisposition for efferocytosis, macrophage differentiation, locomotion, and inflammation [5]. However, the precise molecular mechanisms involved in BM macrophage response to the efferocytosis of apoptotic cancer cells remains to be elucidated.

The majority of solid tumors present in areas of permanent or transient hypoxia due to poor vascularization and blood supply [6]. Hypoxic conditions activate hypoxia-inducible factor (HIF) signaling, which has a crucial role in pro-tumorigenic inflammatory processes via cytokine secretion, reactive oxygen species (ROS) production, and angiogenesis [7]. HIFs are heterodimers consisting of an oxygen labile alpha (α) subunit and a stable beta (β) subunit. There are three isoforms of HIF-α, including HIF-1α, HIF-2α (EPAS1), and HIF-3α (IPAS) [8]. HIF-1α upregulates glycolytic genes, such as phosphoglycerate kinase (PGK) and lactate dehydrogenase A (LDHA), whereas HIF-2α induces the expression of genes related to oxygen supply improvement in hypoxic regions, such as erythropoietin [9]. HIF-1α has been identified as a key regulator of proliferative, invasive, and immunosuppressive mechanisms that favor tumor progression [10,11]. Under hypoxic conditions, HIF-1α hydroxylation by prolyl hydroxylase is reduced. This inhibits the HIF-1α–von Hippel-Lindau (VHL) interaction and consequent HIF-1α degradation by the ubiquitin E3 ligase complex [12]. Therefore, HIF-1α is stabilized in the cytosol and translocated to the nucleus to promote the transcription of multiple target genes [13]. HIF-1α is strikingly upregulated under hypoxic conditions; however, HIF-1α can also be regulated at transcriptional, translational, and post-translational levels under normoxic conditions [14]. HIF-1α is expressed and stabilized in immune cells via hypoxia or other factors, such as inflammation, cancer, and infectious micro-organisms [15,16]. HIF-1α is crucial for myeloid cell-mediated inflammation [17] and it has been demonstrated that tumor-associated macrophages (TAMs) also express HIF-2α under hypoxic conditions [18,19]. Various studies have shown a relationship between HIF-1α induction and signal transducer and activator of transcription 3 (STAT3) activation at transcriptional and post-translational levels [20,21,22,23].

Previous reports have shown that low oxygen concentrations in tumors promote the secretion of cytokines and chemokines that recruit pro-tumorigenic Tregs, tumor-associated macrophages, neutrophils, B cells, and myeloid-derived suppressor cells (MDSCs) to support tumor growth [24,25,26]. One of these cytokines is macrophage migration inhibitory factor (MIF), which is a direct target gene of HIF-1α [27] and a hypoxia-induced gene in colon and breast cancer cells [28,29]. MIF acts as an autocrine or paracrine cytokine, is upregulated in several types of cancer [30,31], and its expression correlates with disease malignancy and invasiveness [32]. Studies have consistently demonstrated that MIF primarily signals through CD74 in association with CD44, CXCR2, CXCR4, and CXCR7 to activate the ERK/MAP kinase cascade [33,34]. Finally, MIF signaling induces the activation and secretion of pro-tumorigenic cytokines to support tumor growth [35,36].

Using single-cell transcriptomic sequencing, we investigated the signature changes in BM macrophage gene expression during the efferocytosis of apoptotic prostate cancer cells. We found that BM macrophages engulfing apoptotic prostate cancer cells promoted HIF-1α stability and subsequent HIF-1α and p-STAT3 nuclear translocation to induce the expression of the pro-inflammatory cytokine MIF. These findings suggest that p-STAT3/HIF1α/MIF signaling in tumor-associated BM macrophages may promote inflammation in the prostate cancer bone microenvironment.

## 2. Materials and Methods

### 2.1. Animals and Cell Lines

All animal experiments were performed with approval from the University of Michigan Institutional Animal Care and Use Committee. Immunocompetent C57BL/6J, FVB/NJ, B6.129P2-Lyz2^tm1(cre)Ifo^/J (LysMCre), B6.129-*Hif1a*^tm3Rsjo^/J (*Hif1a*^flox/flox^) [37], and *Epas1*^tm1Mcs^/J (*Epas1*^flox/flox^) [38] mice were purchased from the Jackson Laboratory (Bar Harbor, ME, USA). The *Hif1a*^flox/flox^ and *Epas1*^flox/flox^ (HIF-2α^flox/flox^) mice were crossed consecutively with LysMCre mice to achieve the *Hif1a*^flox/flox^-LysMCre^+/−^ (*Hif1a*^mut^) and *Epas1*^flox/flox^-LysMCre^+/−^ (*Epas1*^mut^) mice that exhibit HIF-1α and HIF-2α inactivation in myeloid cells, including BM macrophages. *Hif1a*^flox/flox^ and *Epas1*^flox/flox^ mice were used as experimental controls (WT).

RM1 is a (Ras+Myc)-induced prostate cancer cell line developed in C57BL/6J mice and was a gift from Timothy C. Thompson (Baylor College of Medicine, Houston, TX, USA) [39,40]. The Myc-CaP prostate cancer cell line is derived from a prostate carcinoma from Hi-Myc FVB/NJ mice and was donated by Russell Taichman and Frank Cackowski (University of Michigan, Ann Arbor, MI, USA) [41]. Both cell lines were cultured in RPMI 1640 medium containing 10% fetal bovine serum (FBS) and grown at 37 °C with ambient O_2_ and 5% CO_2_.

### 2.2. Murine Efferocytosis In Vitro Model

Bone marrow macrophages (MΦs, in Figures) were isolated from 4–6-week-old male C57BL/6J, FVB/NJ, *Hif1a*^mut^, *Epas1*^mut^, *Hif1a*^flox/flox^, and *Epas1*^flox/flox^ mice via flushing of the femur and tibia with minimum essential medium eagle—alpha modification (αMEM) supplemented with L-glutamine, antibiotic-antimycotic 1×, and 10% fetal bovine serum (FBS). BM macrophages were cultured in αMEM (L-glutamine, antibiotic-antimycotic 1×, 10% FBS) in the presence of macrophage colony stimulating factor (M-CSF) (30 ng/mL, #315-02, Peprotech, Rocky Hill, NJ, USA). After four days in culture, BM macrophages were independently plated in 2 × 10^6^ cells/well with αMEM (L-glutamine, antibiotic-antimycotic 1×, 0.25% FBS) for co-culture experiments. RM1 and Myc-CaP cells were exposed to UV light for 30 min to induce apoptosis. Apoptotic (a) cells (>90% trypan blue incorporation) were co-cultured with BM macrophages at a 1:1 ratio in αMEM (L-glutamine, 0.25% FBS) for 16–18 h. BM macrophages from mutant mice were compared with those from respective littermate controls.

Prolyl hydroxylase inhibition in efferocytic and non-efferocytic BM macrophages was performed using 10µM Roxadustat (FG-4592) (Cayman Chemical, Ann Arbor, MI, USA, 15294) for 16–18 h. STAT3 inhibition in efferocytic and non-efferocytic BM macrophages was performed using 5 µM Stattic (Cayman Chemical, 14590) or 60 µM S3I-201 (Millipore Sigma, Burlington, MA, USA, SML0330) for 10 min and then medium was replaced and BM macrophages were further incubated with apoptotic cancer cells for 2 h. Proteasome inhibition in efferocytic and non-efferocytic BM macrophages was performed using 10 µM MG-132 (Cayman Chemical, 10012628) for 1 h. BM macrophages alone were treated with 200 ng/ml of recombinant MIF protein (#50066-M08H, Sino Biological, Chesterbrook, PA, USA) for either 2 h for Western blot analysis or 8 h for RNA expression analysis.

### 2.3. Single-Cell Library Preparation and RNA Sequencing

A modified murine efferocytosis in vitro model was used. Apoptotic RM1 cells were labeled with CellTrace™ CFSE (#C34554, ThermoFisher Scientific, Waltham, MA, USA) and then co-cultured with BM macrophages for 16–18 h. Efferocytic and non-efferocytic BM macrophages were collected and incubated in fluorescence-activated cell sorter (FACS) staining buffer (phosphate buffered saline-1X, 0.2% bovine serum albumin). F4/80 antibody and isotype control were added and incubated for 30 min at 4 °C. F4/80^+^ only (non-efferocytic BM macrophages) and F4/80^+^CFSE^+^ (efferocytic BM macrophages) were sorted using a BD FACSAriaTM III (BD biosciences, San Jose, CA, USA). Antibody information is available in Appendix A.

The single-cell RNA-sequencing (scRNA-Seq) libraries were prepared at the University of Michigan Advanced Genomics Core using a 10X Genomics Chromium Next GEM Single Cell 3′ Kit v3.1 (part number 1000268) following the manufacturer’s protocol. Cell suspensions were diluted to target a recovery of 10,000 cells per sample. The libraries were run on an Agilent TapeStation 4200 (part number G2991BA) for library quality control before sequencing. Libraries were sequenced at a depth of 50,000 reads/cell on a NovaSeq6000 with the following run configuration: Read 1—150 cycles; i7 index read—8 cycles; Read 2—150 cycles.

### 2.4. Single-Cell RNA-Sequencing Analysis and Visualization

The scRNA-Seq data were processed using 10X Genomics CellRanger software suite v3.0.0. Briefly, fastq files from each of the samples were mapped to the mouse genome mm10 and genes were counted using CellRanger software and STAR aligner [42]. The barcode-gene matrices were further analyzed using the Seurat R package (v3.1) [43]. To remove low-quality cells, cells that expressed less than 200 genes or less than 1000 transcripts or cells that had greater than 10% mitochondrial genes were filtered from the datasets following standard practices (Appendix A). For genes, only the top 5000 variable genes were included for downstream analysis. Samples were then normalized and integrated according to the Seurat-suggested pipeline. To reduce the dimensionality of the samples, we first performed a principal component analysis (PCA). The number of principal components for further downstream applications were 20 and uniform manifold approximation and projection (UMAP) was employed for final dimensionality reduction and visualization of the data.

### 2.5. Differential Expression and Gene Ontology Analysis

Differential expression analysis was conducted using the DESingle R package [44]. Genes with a false discovery rate-adjusted *p*-value < 0.05 were considered differentially expressed. For pathway analysis, we used PANTHER analysis [45,46] with the gene ontology database [47,48]. Only genes differentially expressed and upregulated in the efferocytic BM macrophages in both experiments were included in the GSEA analysis.

### 2.6. Western Blot Analysis, Immunoprecipitation, and Subcellular Fractionation Assays

Whole cell lysates were extracted in Cell Lysis Buffer 1X (#9803, Cell signaling Technology, Danvers, MA, USA) containing 1X protease and phosphatase inhibitor cocktail (#78440, ThermoFisher Scientific, Waltham, MA, USA). Estimation of protein concentration was performed using the Bradford assay (#5000006, BioRad, Hercules, CA, USA). Samples (15 µg each) were diluted using 1X Laemmli Sample Buffer (#1610747, 4X stock, BioRad, Hercules, CA, USA) with 10% β-Mercaptoethanol (#M3148, Millipore Sigma, Burlington, MA, USA). Protein lysates were separated using 4–20% Mini-PROTEAN^®^ TGX Stain-Free™ gels (#4568096, BioRad, Hercules, CA, USA) and transferred to PVDF membrane using the Trans-Blot Turbo RTA kit (#1704272, BioRad, Hercules, CA, USA). The membrane was blocked with 5% milk in 1X TBS with 0.1% Tween for 1 h at room temperature and then incubated with primary antibodies in 5% BSA overnight at 4 °C. Secondary antibody was diluted in 5% milk in 1X TBS with 0.1% Tween. For immunoprecipitation assays, macrophage and apoptotic prostate cancer cell co-cultures were lysed on ice with 1% Triton X-100 in 1X PBS with a 1X protease and phosphatase inhibitor cocktail. Whole cell lysates were immunoprecipitated using anti-HIF-1α rabbit antibody and protein A-magnetic beads (#73778, Cell Signaling Technology, Danvers, MA, USA) during an overnight incubation at 4 °C. Binding and washing were performed in the same lysis buffer. HIF-1α immunocomplex was resuspended in Laemmli sample buffer (30 µL), and 15 µL was processed for immunoblotting with anti-ubiquitin mouse antibody. For subcellular fractionation assays efferocytic and non-efferocytic BM macrophages were collected and processed using a Cell Fractionation Kit, following the manufacture’s protocol (#9038, Cell Signaling Technology, Danvers, MA, USA). Blots were developed using SuperSignal™ West Femto Maximum Sensitivity Substrate (#34095, ThermoFisher Scientific, Waltham, MA, USA). Protein gels used for protein normalization and blots were imaged using the ChemiDoc™ MP Imaging System (#12003154, BioRad, Hercules, CA, USA). Antibody information is available in Appendix A.

### 2.7. Efferocytic and Non-Efferocytic BM Macrophage Isolation and Culture

Apoptotic RM1 cells were labeled with CellTracker™ DeepRed Dye (#C34565, ThermoFisher Scientific, Waltham, MA, USA) and then co-cultured with BM macrophages for 16 h. Efferocytic and non-efferocytic BM macrophages were collected in fluorescence-activated cell sorter (FACS) staining buffer (phosphate buffered saline-1X, 0.2% bovine serum albumin) using F4/80-FITC antibody and isotype control for 30 min at 4 °C. F4/80^+^ non-efferocytic BM macrophages and F4/80^+^DeepRed^+^ efferocytic BM macrophages (engulfing RM1) were sorted using a BD FACSAriaTM III (BD bio-sciences, San Jose, CA, USA). Antibody information is available in Appendix A. Efferocytic and non-efferocytic BM macrophages from each mouse were quantified and plated in 8 × 10^4^ cells per well in a 96-well plate for 6 h in 150 µL of low-serum medium (0.25% FBS) containing M-CSF (as described in 2.2). The culture medium from each well was collected and centrifuged at 1500 rpm for 10 min to separate from any floating cell and 12 µL of medium from each sample was analyzed by Western blotting, as described in 2.6.

### 2.8. RT-qPCR

Cells were harvested using an RNeasy Mini Kit (#74106, Qiagen, Germantown, MD, USA) and RNA was eluted with nuclease-free water and then quantified using a NanoDrop 2000 (Thermo Scientific, Waltham, MA, USA). cDNA was synthetized with 1µg of RNA in a 20 µL reaction mixture using a High-Capacity cDNA Reverse Transcription Kit (#4368814, ThermoFisher Scientific, Waltham, MA, USA). RT-qPCR was performed using TaqMan^®^ probes and Gene Expression qPCR Assays TaqMan Gene Express (#4369016, ThermoFisher Scientific, Waltham, MA, USA) with 40 cycles on an ABI PRISM 7700 (Applied Biosystems, Waltham, MA, USA, USA). The analysis was performed using the 2^−ΔΔCT^ method [49]. TaqMan^®^ probe information is available in Appendix A.

### 2.9. Statistics

All experiments with BM macrophages were obtained from at least 3 independent mice per group. Results were normalized to the BM macrophage-only control group and data from independent experiments were pooled together. Statistical analyses were performed using GraphPad Prism 9 (GraphPad Software, version 9.1.0, San Diego, CA, USA) using ordinary and repeated measures one-way analysis of variance (ANOVA) with Tukey’s multiple comparison and unpaired *t*-test analyses, with a significance of *p* < 0.05. No power analysis was performed. No outliers’ results were excluded in any experiment. The code used for the scRNA-Seq analysis is included in the ‘submission_code_log.R’ file. The file ‘DE_genes_roca45_and_roca67.csv’ contains the full list of results for the differential gene expression analysis for both experiments.

Details for experiments and number of independent mice used can be found in the figure’s description. The specifics of the statistical methods used are detailed in these sections.

## 3. Results

### 3.1. Single-Cell Analyses of BM Macrophages Engulfing Apoptotic Prostate Cancer Cells Showed a Distinct Transcriptional Signature and Activation of Hypoxia-Related Genes

Published findings suggest that macrophages induce distinctive tumor-promoting signaling in response to the efferocytosis of apoptotic cancer cells [3,4,50,51]. However, the mechanisms that govern these specific responses in connection to tumor acceleration are not completely understood. To further investigate efferocytosis-mediated signaling in macrophages, primary BM macrophages from immunocompetent C57BL/6J mice (~90% F4/80^+^, as shown in Appendix B—Figure A6) were co-cultured with CFSE^+^ (pre-labeled) apoptotic prostate cancer RM1 cells. BM macrophages engulfing (efferocytic) these apoptotic cancer cells were compared to non-engulfing (non-efferocytic) BM macrophages by scRNA-Seq upon sorting by flow cytometry (Figure 1A). After the identification of high quality sequenced single cells (Appendix B—Table A1), UMAP [52] was applied for the dimension reduction of single-cell analysis and visualization of the transcriptional data for efferocytic and non-efferocytic BM macrophages. As shown in Figure 1B, the cell distribution in two UMAP projections depicts differential cluster enrichments in efferocytic vs. non-efferocytic BM macrophages from two independent experiments (Exp. 1 and Exp. 2). For example, efferocytic BM macrophages showed an enriched cluster in the direction of increased UMAP-2-projection while the opposite is observed in non-efferocytic BM macrophages, shown in the split visualization of these cells (Figure 1B). These results correlate with distinct transcriptional heatmaps in efferocytic relative to non-efferocytic BM macrophages (Figure 1C), where the great majority of differentially expressed genes (DEGs) were significantly changed in the same direction (3277 vs. 482) in both experiments, as visualized in the Venn diagram in Figure 1D. DEGs commonly upregulated in efferocytic BM macrophages in both experiments were further processed using the PANTHER analysis [45,46] and a gene ontology (GO) database [47,48] to identify the relevant biological pathways. Among the identified significantly enriched GO-biological processes, we found pathways related to the innate immune system and wound healing responses, which are related to the phagocytosis of apoptotic cells, inflammation, and regeneration (Figure 1E, list of enriched GO terms in Appendix B—Table A1) [53,54]. Intriguingly, biological processes related to hypoxia were identified even though these experiments were performed under normoxia (normal oxygen conditions), which suggests that efferocytosis-mediated activation and upregulation of factors directly related to cellular hypoxia is independent of the oxygen concentration (Figure 1E,F).

### 3.2. Efferocytosis of Apoptotic Cancer Cells Stabilized HIF-1α in BM Macrophages and Is Mediated by the Activation of STAT3

Single-cell analysis of efferocytic BM macrophages identified upregulated molecules associated with the “cellular hypoxia” gene ontology (GO) term (Figure 1E). To investigate these findings in the overall macrophage population, co-cultures of BM macrophages with apoptotic prostate cancer RM1 cells (workflow Figure 1A, without sorting) were analyzed. Selected hypoxia-GO-associated genes showing significant upregulation by sc-RNAseq were investigated by RT-qPCR from total RNA isolated from co-cultures of independent BM populations. Fold changes in efferocytic relative to control non-efferocytic BM macrophages were calculated and plotted in Figure 2A. The majority of analyzed genes showed a significant mRNA increase in efferocytic BM macrophages relative to the control, corroborating the sc-RNAseq results. Some of these molecules are crucial components of glycolysis, including *Pdk1*, *Pgk1*, and *Ldha,* and are known targets of hypoxia-inducible factor 1a (HIF-1α), a master transcriptional regulator of the cellular response to hypoxia that promotes a metabolic switch to glycolysis [21,55]. *Hif1a* mRNA was upregulated in the overall population of efferocytic BM macrophages (although it was not identified as an upregulated gene in the single-cell experiments). Contrary to the results observed by single-cell experiments, the hypoxia-inducible transcription factor *Epas1* (HIF-2α) [18] showed a small but significant decrease in mRNA expression via efferocytosis.

Because HIF-1α is largely regulated post-transcriptionally resulting in a protein targeted for degradation under normoxic conditions [12,56], HIF-1α protein was further evaluated by Western blot analysis. As shown in Figure 2B, Western blot analysis of efferocytic BM macrophages (co-cultured for 16–18 h with apoptotic RM1 cells) evidenced a significant increase in HIF-1α induced by efferocytosis. These findings were corroborated in efferocytic BM macrophages co-cultured with murine prostate cancer Myc-CaP cells, which share several molecular characteristics of human prostate cancer [57,58]. As Myc-CaP cancer cells were obtained from FVB/NJ mice, primary macrophages were obtained from the bone marrow of the same strain. Similar results were observed using this model (Figure 2C). These results suggest that HIF-1α is stabilized in BM macrophages engulfing apoptotic cancer cells.

The potential mechanism inducing HIF-1α stabilization was further investigated. Previous findings suggested that one potential mechanism leading to HIF-1α stabilization is interaction with activated (phosphorylated) STAT3 [20,22,23,59]. Moreover, nuclear cooperative translocation of HIF-1α and p-STAT3 has been observed in the tumor microenvironment [60,61]. Since STAT3 activation is sustained in efferocytic BM macrophages and considered a hallmark macrophage response to engulfing apoptotic cancer cells, it was hypothesized that STAT3 activation (phosphorylation at Tyr705) is critical in HIF-1α stabilization by efferocytosis. We investigated this hypothesis using two well-characterized STAT3 phosphorylation inhibitors: Stattic and S3I-201 [62,63]. Both inhibitors significantly reduced the activation of STAT3 after a short treatment of BM macrophages, followed by 2 h of incubation with apoptotic cancer cells. This treatment also impacted the stabilization of HIF-1α (Figure 2D–G and Appendix B—Figure A1). Similarly, these findings were observed in BM macrophage efferocytosis of apoptotic Myc-CaP cells (Appendix B—Figure A2). These results strongly support the hypothesis that STAT3 activation is a critical signal that mediates the stabilization of HIF-1α via efferocytosis.

To assess how HIF-1α ubiquitination (Ub) levels are affected by the ubiquitin-mediated proteasomal degradation pathway, HIF-1α immunoprecipitation assays were performed using protein lysates from BM macrophages co-cultured with apoptotic prostate cancer RM1 cells vs. control. The immunoprecipitated samples were analyzed by Western blotting using a ubiquitin-specific antibody. The blot showed reduced ubiquitin levels in efferocytic vs. non-efferocytic BM macrophages after normalization to immunoprecipitated HIF-1α protein. This reduction was statistically significant when BM macrophages were treated with MG-132, a degradation inhibitor of ubiquitin-conjugated proteins (Ub-HIF-1α) (Figure 2H). To further investigate these findings, cytoplasmic and nuclear fractions were isolated from efferocytic and non-efferocytic BM macrophages. Protein lysates from each fraction were analyzed by Western blotting and the results showed a higher expression of HIF-1α and p-STAT3 in the nuclear factions of efferocytic BM macrophages (Figure 2I). Histone H3 immunoblot was used as a nuclear marker to demonstrate the efficacy of the nuclear fractionation. These results suggest that macrophage efferocytosis promotes HIF-1α stabilization along with co-activation of p-STAT3.

### 3.3. Efferocytosis of Apoptotic Cancer Cells Stimulated the Expression of Pro-Inflammatory MIF Cytokine in BM Macrophages

Accumulating experimental evidence suggests that efferocytosis of apoptotic cancer cells accelerates tumor progression and metastatic growth by fostering an inflammatory and immunosuppressive microenvironment [64,65]. Single-cell data identified the negative regulation of the immune system process (GO: 0002683; related to the immunosuppressive response) as one of the GO terms upregulated in efferocytic BM macrophages. Using STRING, a database of known and predicted protein–protein interactions, we identified a strong network association between this immune response and the biological process of hypoxia (GO: 0071456) (Figure 3A, GO gene list in Appendix B—Table A2 and Table A3). Although not identified by single-cell analysis, STAT3 was added because of its key role in the stabilization of HIF-1α, as shown in Figure 2. Central nodes identified in this network are HIF-1α, Myc, and STAT3 and the findings show direct or indirect interactions between hypoxia and the negative immune regulation processes. Furthermore, single-cell analysis identified the cytokine macrophage migration inhibitory factor *Mif* as upregulated in efferocytic BM macrophages (*p* < 10^−6^) (Figure 1F and Figure 3B and Appendix B—Figure A5). *Mif* belongs to both GO: 0002683 and GO: 0071456 [48] and mediates both immunosuppression and inflammation and has been associated with increased tumorigenesis and disease progression in different cancer types including prostate cancer [66].

MIF changes were investigated in the overall macrophage population co-cultured with apoptotic RM1 prostate cancer cells by RT-qPCR and Western blot analysis. In correlation with single-cell results, both *Mif* mRNA (Figure 3C) and MIF protein increased in BM macrophage efferocytosis of apoptotic RM1 cells (Figure 3D). Similarly, BM macrophages isolated from FVB mice upregulated MIF protein upon the efferocytosis of Myc-CaP prostate cancer cells (Figure 3E). Secreted MIF was further analyzed in the efferocytic BM macrophages and compared to the control BM macrophages. To differentiate between the newly secreted endogenous MIF and the potential MIF present in the media from cancer cells, BM macrophages engulfing apoptotic RM1 (efferocytic) from three independent BM macrophage co-cultures were sorted by flow cytometry (double positive F4/80-FITC^+^/RM1(a)-DeepRed^+^) and compared to F4/80^+^ non-engulfing BM macrophages (non-efferocytic, also sorted by flow cytometry) (Figure 3F). Both populations were cultured for 6 h and the conditioned media from each sample were analyzed by Western blotting. The analysis revealed a significant increase in secreted MIF detected in the conditioned media from efferocytic relative to non-efferocytic BM macrophages (Figure 3G). These results indicate that efferocytosis induces MIF secretion by BM macrophages to the extracellular microenvironment.

Altogether, these findings indicate MIF is part of the signaling response of BM macrophages to the engulfing of apoptotic prostate cancer cells and suggests a network connection with the activation of hypoxia-related molecules by efferocytosis.

To investigate if the pro-inflammatory response is induced by BM macrophage efferocytosis of apoptotic cancer cells and not by apoptotic normal epithelial cells, HIF-1α stabilization, STAT3 phosphorylation, and MIF expression were assessed in BM macrophages co-cultured with apoptotic RM1 prostate cancer cells and compared to BM macrophages co-cultured with apoptotic mPEC normal prostate epithelial cells. Western blot analysis showed that HIF-1α is highly stable in RM1-efferocytic BM macrophages and this is significantly increased in relation to mPEC-efferocytic BM macrophages (Figure 4A). Moreover, while STAT3 phosphorylation was increased in both RM1- and mPEC-efferocytic BM macrophages, p-STAT3 levels were significantly higher in BM macrophage efferocytosis of apoptotic RM1 cancer cells vs. mPECs (Figure 4B). Finally, MIF expression was significantly upregulated only in RM1-efferocytic BM macrophages (Figure 4C).

These data confirm that BM macrophage efferocytosis of apoptotic cancer cells promotes a unique pro-inflammatory response and that this feature may contribute to cancer cell growth in the tumor microenvironment.

### 3.4. HIF-1α Mediated the Expression of MIF Cytokine in Efferocytic BM Macrophages

The implication of HIF-1α in tumor-promoting inflammation, immunosuppression, and metastasis has been documented in different cancer models in relation to hypoxia [7]. We hypothesized that the stabilization of HIF-1α in BM macrophages by the clearance of apoptotic cancer cells induces the expression of key pro-inflammatory cytokines. This was investigated by crossing LysMCre mice with *Hif1a*^flox/flox^ mice to obtain mutated HIF-1α myeloid lineage-mutant mice (*Hif1a*^mut^). These mice have a null allele in the Cre-expressing cells (myeloid) that lacks the exon 2 of *Hif1a* and were used to obtain *Hif1a*^mut^ BM macrophages.

WT and *Hif1a*^mut^ BM macrophages were characterized by flow cytometry. No changes were observed in the BM macrophage expression profile of surface marker F4/80 (Appendix B—Figure A6). In addition, the efferocytic capability (measured by the percent of engulfment of apoptotic RM1) was no different between WT and *Hif1a*^mut^ BM macrophages (Appendix B—Figure A7). Efferocytosis induced a significant increase in the CD206^high^ population in WT BM macrophages, suggesting increased M2-like polarization (Appendix B—Figure A8). This trend increase (although not significant; *p* = 0.0516) was also observed in *Hif1a*^mut^ BM macrophages. However, *Hif1a*^mut^ BM macrophages alone also showed a slight increase in the CD206^high^ population when compared to WT, suggesting a shift in M2-like polarization induced by the loss of HIF-1α function in BM macrophages. However, no differences were found in CD86, an M1-marker, in these BM macrophages.

Relative *Hif1a* gene expression was quantified with a probe that specifically targets *Hif1a* exon 2. RT-qPCR analysis showed the upregulation of *Hif1a* mRNA in the efferocytic WT BM macrophages relative to the controls (Figure 5A). A significant decrease in the *Hif1a* mRNA containing the exon 2 was observed in *Hif1a*^mut^ BM macrophages relative to the control WT BM macrophages, with no efferocytic response noted (Figure 5A). In addition, the characterization of *Hif1a*^mut^ BM macrophages by Western blot analysis demonstrated a lower molecular weight of HIF-1α in *Hif1a*^mut^, which corresponds with the deletion of the DNA binding domain encoded by the exon 2 (Figure 5C) by Cre-induced recombination and renders a non-functional *Hif1a*^mut^ protein. However, even this mutant protein was stabilized by efferocytosis as shown by quantitative Western blot analyses, which suggests that STAT3-mediated stabilization is independent of HIF-1α binding to the hypoxia response element (HRE) DNA (Figure 5C). Notably, STAT3 activation remained unaffected in *Hif1a*^mut^ BM macrophages (Appendix B—Figure A9).

Previous studies suggest a link between HIF-1α and MIF in different cell models, including macrophages [67,68]. We investigated the expression of MIF and other pro-inflammatory factors as potential targets of HIF-1α. These included critical inflammatory cytokines previously found upregulated in efferocytic BM macrophages: CXCL1, CXCL5, IL6, and CXCL4 (also known as platelet factor 4, *Pf4*) [69]. From the selected cytokines, it was found that *Mif* and *Cxcl4* expression was significantly reduced in *Hif1a*^mut^ efferocytic BM macrophages relative to the wild type (WT) after normalization to their respective control (non-efferocytic BM macrophages) (Figure 5B). There was a significant reduction in MIF protein expression and no upregulation was observed by efferocytosis in the *Hif1a*^mut^ BM macrophages relative to WT (Figure 5D). In contrast, upregulation of MIF by efferocytosis was found in WT BM macrophages as previously shown in Figure 3D–E.

To address the specificity of HIF-1α in the control of MIF regulation, similar experiments were performed with *Epas1*-myeloid lineage-mutant (*Epas1*^mut^) mice. These mice were obtained by crossing LysMCre mice with *Epas1*^flox/flox^ mice. Mutant mice expressed significantly lower levels of *Epas1* mRNA by RT-qPCR using the primer/probe set corresponding to the deleted exon2 (Appendix B—Figure A10A). RT-qPCR analysis of pro-inflammatory cytokines showed a significant decrease in *Cxcl1* in the *Epas1*^mut^ mice, with no changes in *Mif* and *Cxcl4* (Appendix B—Figure A10B), which differs from the observed HIF-1α-mediated regulation in BM macrophages (Figure 5B). Quantitative protein analysis of *Epas1*^mut^ BM macrophages revealed no change by efferocytosis in HIF-1α stabilization in the *Epas1*^mut^ BM macrophages relative to WT control (Appendix B—Figure A10C), nor in MIF expression when comparing *Epas1*^mut^ vs. WT (Appendix B—Figure A10D). Furthermore, relative to the control *Epas1*^mut^ BM macrophages, efferocytic *Epas1*^mut^ BM macrophages showed a significant increase in MIF (Appendix B—Figure A10D), while no change was observed in efferocytic *Hif1a*^mut^ BM macrophages (Figure 5D). These results highlight the specificity in the regulation of MIF expression by HIF-1 relative to EPAS1, a similar hypoxia-inducible transcription factor.

HIF-1α-mediated MIF regulation was further investigated by using a HIF-prolyl-hydroxylase inhibitor FG-4592 (also known as Roxadustat) [70]. FG-4592 was used in efferocytosis assays and a strong correlation was observed between the stabilization of HIF-1α and MIF protein expression. FG-4592 alone stabilized HIF-1α, revealed by the upregulation of MIF protein in non-efferocytic BM macrophages. Intriguingly, when the inhibitor was used in efferocytic BM macrophages a further increase in HIF-1α and MIF protein was observed (Figure 6A). RT-qPCR analysis showed a significant increase in *Mif* mRNA induced by FG-4592 relative to the control; however, no further increase was observed in efferocytic BM macrophages (Figure 6B).

Altogether, these findings suggest that HIF-1α mediates the expression of MIF, where HIF-1α stabilization via efferocytosis or prolyl-hydroxylase inhibitor significantly upregulates MIF expression in BM macrophages.

### 3.5. MIF Activateds Inflammation in BM Macrophages

CD74 is a critical receptor for MIF signal transduction in cells; however, CD74 lacks kinase activity and requires a complex formation with other co-receptors, including CD44 and CXCR4 [33,71,72]. Results from single-cell data analyses identified a significant downregulation of CD74 in efferocytic compared to non-efferocytic BM macrophages (Figure 7A, Appendix B—Figure A11), while no significant differences were detected in the co-receptors CD44 or CXCR4. The downregulation of CD74 was also evident in the overall efferocytic BM macrophages relative to the control, demonstrated by RT-qPCR analyses (Figure 7B).

Although these results do not rule out a potential endocrine signaling, they suggest that MIF secreted from efferocytic BM macrophages (Figure 3F) could induce potent paracrine signaling in non-efferocytic BM macrophages and other cells. To evaluate the MIF-induced signaling in BM macrophages, a purified recombinant MIF protein expressed in mammalian cells was used. BM macrophages (from C57BL/6J and FVB/NJ mice) were incubated for 2 h with MIF and then signaling activation was analyzed by Western blotting with specific phospho-peptide antibodies. A hallmark the MIF transducing signal, resulting in the activation (sustained phosphorylation) of the extracellular signal-related kinase ERK1/2 MAPK [66,73], was found highly upregulated in BM macrophages treated with recombinant MIF (Figure 7C). Furthermore, an increase in the critical inflammatory NF-kB signaling (phospho-p65) was observed in BM macrophages treated with MIF (Figure 7C).

This potent inflammation-transduced signaling pathway was further correlated with the increased expression of several pro-inflammatory factors in MIF-activated BM macrophages (from C57BL/6J), including: *Ccl5*, *Cxcl5*, *Il6*, *Cxcl1*, *Cxcl4, IL1b, and Tnf,* as well as the glucose transporter *Glut1* (Figure 7D). Previous studies have demonstrated that these cytokines mediate a pro-inflammatory macrophage activation in different environments [74,75,76]. Similar results were observed in FVB BM macrophages, where MIF activated the expression of pro-inflammatory factors, except for *Cxcl4* and the *Glut1* transporter (Figure 7F). In contrast, from the tested anti-inflammatory signatures, only CD36 was upregulated in C57BL/6J BM macrophages, while CD206 was downregulated in the FVB BM macrophages. The activation of pro-inflammatory cytokines has also been detected in BM macrophages engulfing prostate cancer cells and functions to accelerate tumor growth in bone, as previously shown [2]. Altogether, these results suggest that STAT3–HIF-1α–MIF is a potent signaling axis induced by BM macrophage efferocytosis of apoptotic cancer cells, which may act via paracrine signaling to induce macrophage-mediated inflammation in the bone tumor milieu.

## 4. Discussion

Inflammation has a major impact on cancer progression and metastasis in various organs [77,78,79]. One of the inflammatory mechanisms that promotes tumor growth is the secretion of cytokines and chemokines by cancer and immune cells in the tumor microenvironment [80,81]. Tumor-associated macrophages play a critical role in accelerating tumor progression in different cancer types [82,83]. In bone, apoptotic cancer cell efferocytosis by macrophages generates a unique inflammatory milieu rich in cytokines, which promote and support tumor progression [3,4]. Here, the molecular mechanisms that induce this tumor-promoting inflammatory response in bone marrow macrophages during the efferocytosis of apoptotic prostate cancer cells were investigated.

Analysis of the single-cell transcriptomics of sorted BM macrophages engulfing apoptotic prostate cancer RM1 cells (efferocytic, F4/80^+^CFSE^+^) vs. sorted BM macrophages alone (non-efferocytic, F4/80^+^) by flow cytometry (Figure 1A) revealed two distinctive clusters of cells, each with a unique gene expression profile. GO term enrichment analysis of the upregulated genes in efferocytic BM macrophages identified several biological pathways including those directly related to macrophage functions, such as “response to wounding”, “innate immune response”, and “regulation of vasculature development” [84]. Although the efferocytosis experiments in vitro were conducted under normoxic levels, the analysis identified GO terms related to “cellular responses to hypoxia” or “decreased oxygen levels” in efferocytic BM macrophages, these terms included genes that are known targets of HIF-1, a master transcriptional regulator of the cellular response to hypoxia [85]. Although HIF-1α is known to be critically regulated by oxygen-dependent hydroxylation leading to ubiquitination and subsequent proteasome degradation [13], several factors could induce its stabilization under normoxic conditions. For example, in macrophages activated by inflammatory stimuli and in tumor-associated macrophages, normoxic stabilization of HIF-1α may be induced by metabolites, including succinate and lactate [86,87]. In cancer cells, HIF-1α may also be regulated by inducing high transcriptional and translational activity via different pathways: MAPK/ERK, JAK/STAT, and PI3K/AKT/mTOR [88]. The hypoxia-independent expression of HIF-1α has been observed in prostate cancer tumors, in correlation with recurrence following surgery or therapy, increased chemoresistance, and accelerated metastatic progression, suggesting that alternative mechanisms of post-translational stabilization could lead to its accumulation and transcriptional activity in non-hypoxic environments [89,90]. Here, we found that efferocytic BM macrophages promoted HIF-1α stabilization and induced a strong and sustained phosphorylation of STAT3 under normoxic conditions. A short treatment with two specific STAT3 inhibitors significantly reduced HIF-1α in BM macrophages after a 2 hr incubation with apoptotic cancer cells; however, HIF-1α stabilization was not completely abrogated. A plausible explanation is that the short treatment of inhibitors on BM macrophages and then their removal from culture (to prevent side effects) may have resulted in the reactivation of STAT3 signaling. As shown by Western blot analysis, STAT3 is not completely inactivated. A proposed mechanism suggests that activated STAT3 interacts with HIF-1α to inhibit von Hippel-Lindau (VHL) binding to HIF-1α, leading to decreased ubiquitination and stabilization [22]. However, these experiments were performed by overexpressing STAT3 and, in our hands, we attempted without success to show a direct interaction between endogenous p-STAT3 and HIF-1α. Nevertheless, by blocking HIF-1α proteasomal degradation, the immunoprecipitation experiments demonstrated a reduced HIF-1α ubiquitination via efferocytosis, which correlates with HIF-1α accumulation in efferocytic BM macrophages. Altogether, these results suggest that the stabilization of HIF-1α is mediated by p-STAT3 in efferocytic BM macrophages. This stabilization leads to a subsequent nuclear translocation, along with p-STAT3, which correlates with previous studies [60,61]. However, other mechanisms (described above) acting in parallel with STAT3 may also contribute to HIF-1α stabilization.

Previous reports have associated the expression of HIF-1α and its target genes with immunosuppressive functions in the tumor microenvironment [91,92]. Here, a single-cell analyses of efferocytic BM macrophages identified a strong protein network association between the genes related to “cellular response to hypoxia” and “negative regulation of the immune response”, suggesting that HIF-1α signaling in efferocytic BM macrophages may exert immunosuppressive functions in the tumor microenvironment. It has been reported that HIF-1α promotes the secretion of cytokines and chemokines, such as CXC motif chemokine ligand 5 (CXCL5), CXC motif chemokine ligand 12 (CXCL12), chemokine ligand 28 (CCL28), and macrophage migration inhibitory factor (MIF) [24,25,26,93]. One of the genes included in the network analysis was *Mif*. MIF is a pro-inflammatory cytokine expressed by monocytes, macrophages, blood dendritic cells, B cells, neutrophils, eosinophils, mast cells, and basophils and its expression is involved in both innate and adaptive immune processes, as well as in response to hypoxia. Several studies have shown that MIF mediates inflammatory processes, such as sepsis and cancer [94,95]. We demonstrated that *Mif* mRNA as well as intracellular and secreted MIF protein levels were increased in BM macrophages upon the efferocytosis of apoptotic cancer cells. It was found that HIF-1α but not HIF-2α (EPAS1) depletion in BM macrophages reduced the expression of the pro-tumorigenic inflammatory cytokines *Mif* and *Cxcl4* after being exposed to apoptotic prostate cancer cells when compared to WT BM macrophages. In addition, HIF stabilization by the prolyl-hydroxylase inhibitor Roxadustad further stabilized HIF-1α and induced higher MIF protein expression in efferocytic BM macrophages. However, the upregulation of MIF protein did not completely reflect the *Mif* mRNA where no significant differences in the inhibitor-treated efferocytic versus control BM macrophages were found. This discrepancy may be explained by additional MIF stabilization due to intracellular protein–protein interactions, as suggested in previous studies [27,96,97].

Furthermore, it was found that BM macrophage efferocytosis of apoptotic cancer cells promotes higher HIF-1α stabilization when compared to apoptotic normal prostate epithelial cells. In correspondence with the identified mechanism, the efferocytosis of apoptotic prostate cancer cells induced a more robust activation of STAT3 and MIF expression compared to the efferocytosis of apoptotic normal cells. Such strong STAT3 signaling is required to induce a significant stabilization of HIF-1α. However, how apoptotic cancer cells activate significant STAT3 signaling in BM macrophages and how this signal is maintained remains to be elucidated where a potential amplification loop could be critical. Altogether, these findings suggest that HIF-1α specifically regulates MIF expression in efferocytic BM macrophages.

Studies in cancer show that MIF supports tumor progression via different mechanisms. For example, HIF-1α regulates MIF secretion in breast cancer cells to promote tumor proliferation, angiogenesis, and metastasis [29]. In colorectal cancer, it was shown that MIF promoted macrophage recruitment and angiogenesis to accelerate tumor progression [96]. In a model of breast cancer, elevated MIF expression supported tumor growth while a loss of MIF promoted the anti-tumor immune infiltration of CD4^+^/CD8^+^ T cells producing IFNγ, which supported the MIF immunosuppressive function [98]. In other studies, in head and neck squamous carcinoma, the HIF-1α–MIF axis contributed to the recruitment of myeloid (CD11b^+^-Gr-1^+^) cells to enhance tumor growth and angiogenesis [26]. In prostate cancer patients, MIF expression is highly elevated, which has been associated with higher severity and poor outcome [99].

MIF signals through its ligand binding receptor CD74 and its co-receptors (signal transducers) CD44, CXCR2, or CXCR4 [33,71,72]. MIF/CD74 activity promotes immunosuppressive signaling in macrophages and dendritic cells and inhibition of this signaling re-establishes the antitumor immune response in metastatic melanoma [100,101]. Moreover, MIF/CD74 signaling also activates the NF-κB signaling pathway in chronic lymphocytic leukemia [102,103]. CD74/CD44 activation by MIF is followed by the phosphorylation of the proto-oncogene tyrosine-protein kinase (SRC), extracellular signal-related kinase 1/2 (ERK1/2), phosphoinositide 3-kinase (PI3K), and protein kinase B (AKT) [33,71,104,105]. These kinases promote the activation of transcription factors, such as nuclear factor-kappa B (NF-κB, p65), which induces the secretion of pro-inflammatory cytokines, such as IL-6, IL-8, CCL2, and CCL5 [102,106,107,108,109]. Here, we found that *Cd74* expression is downregulated in efferocytic BM macrophages, suggesting potential paracrine signaling in non-efferocytic BM macrophages. Recombinant MIF protein treatment of non-efferocytic BM macrophages activated the ERK1/2 and the p65 pathways and increased the expression of pro-inflammatory factors, such as *Il1b*, *Tnf*, *Cxcl1*, *Cxcl5*, *Il6,* and *Ccl5*. Furthermore, it was found that MIF increased STAT3 activation and HIF-1α protein (Appendix B—Figure A12A,B). However, no differences were observed in *Hif1a* nor in *Mif* gene expression (Appendix B—Figure A12C) evaluated 6 h after the detection of HIF-1α protein (Appendix B—Figure A12B), which suggests that recombinant MIF treatment may induce a transient stabilization of HIF-1α that is insufficient to increase *Mif* gene expression in BM macrophages.

In B cells, a mechanism has been elucidated where the binding of MIF to CD74 initiates its intramembrane cleavage and generates a 42aa domain that binds to cytoplasmic p65 and facilitates its transport to the nucleus to activate transcriptional activity [110]. A similar mechanism could be activated in macrophages; however, to our knowledge, it has yet to be investigated. Such as in the MIF-induced activation of NF-κB (p65) and the expression of potent pro-inflammatory factors in BM macrophages, the observed downregulation of *Cd74* by efferocytosis could function to restrain the autocrine MIF-induced inflammatory response. For example, no mRNA expression differences between CXCL1, CXCL5, and IL-6 were detected in BM macrophages with depleted HIF-1α (*Hif1a*^mut^) relative to WT.

Altogether, these findings reveal a new regulatory mechanism of HIF-1α in BM macrophages during the efferocytosis of apoptotic prostate cancer cells where the p-STAT3/HIF-1α/MIF signaling pathway induces inflammation, a mechanism that would be activated in the bone tumor microenvironment comprising a significant number of apoptotic cancer cells.

## Figures and Tables

**Figure 1 cells-11-03712-f001:**
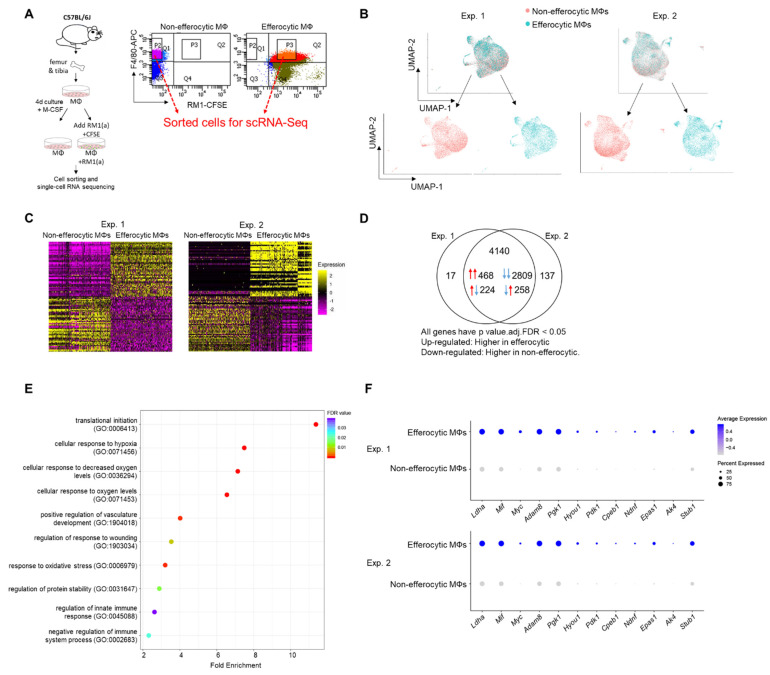
Single-cell experiments comparing efferocytic BM macrophages (engulfing apoptotic cancer cells) vs. control (non-engulfing BM macrophages). (**A**) BM macrophages (MΦ) were isolated from 4 wk old C57BL/6J mice and co-cultured with apoptotic RM1(a) cells (CFSE labeled) for 16–18 h. Efferocytic (F4/80^+^CFSE^+^) and control BM macrophages (F4/80^+^) were sorted and used for single-cell libraries followed by scRNA-Seq. (**B**) UMAP of all cells analyzed (blue: efferocytic BM macrophages; red: non-efferocytic BM macrophages). (**C**) Heatmap of the most differentially expressed genes in either experiment. (**D**) Venn diagram of all differentially expressed genes and their overlap between both experiments. (**E**) Cleveland plot of the most enriched pathways in efferocytic BM macrophages. (**F**) Dot plots of hypoxia-related genes in both experiments. Size relates to the percentage of macrophage cells expressed each gene. Color denotes the average expression for each gene across all expressing cells. Additional results are shown in Appendix B—Table A1, Table A2 and Table A3.

**Figure 2 cells-11-03712-f002:**
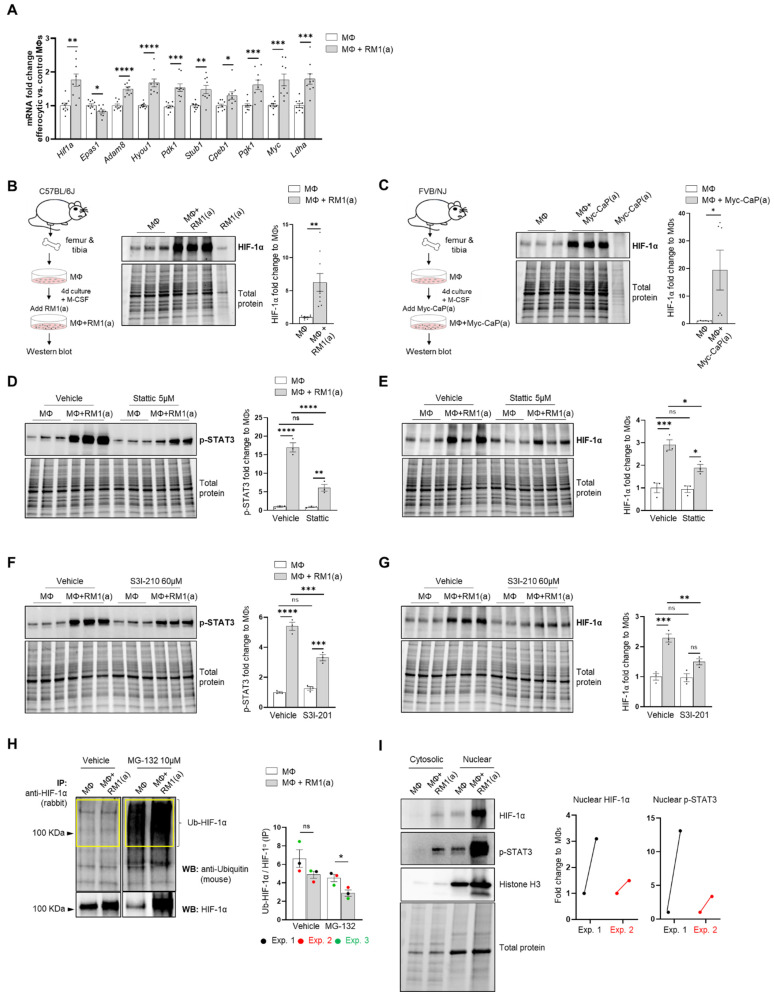
BM macrophage efferocytosis of apoptotic cancer cells promotes HIF-1α stabilization through STAT3 activation. BM macrophages (MΦ) were isolated from 4 wk old C57BL/6J or FVB/NJ mice and co-cultured with apoptotic RM1(a) or Myc-CaP(a) cells for 16–18 h. (**A**) mRNAs isolated from efferocytic and control BM macrophages were analyzed by real time quantitative PCR (RT-qPCR) for a set of genes involved in the cellular response to hypoxia (*n* = 9/group). (**B**,**C**) Protein lysates from C57BL/6J (*n* = 9) and FVB/NJ (*n* = 6) efferocytic BM macrophages were analyzed by Western blotting using HIF-1α antibody. Protein lysates from C57BL/6J efferocytic and control BM macrophages (*n* = 3) treated with 5 µM Stattic and 60 µM S3I-201 STAT3 phosphorylation inhibitors were analyzed by Western blotting for 2 h using (**D**,**F**) p-STAT3 antibody and (**E**,**G**) HIF-1α antibody. (**H**) Cell lysates from efferocytic and non-efferocytic BM macrophages were immunoprecipitated (IP) with HIF-1α antibody and immunoblotted with ubiquitin antibody. The graph depicts the calculated signal of Ub-HIF-1α (ubiquitinated HIF-1α) to total HIF-1α (IP). Ub-HIF-1α signal was measured in the yellow box around the corresponding HIF-1α molecular weight from independent experiments (*n* = 3). (**I**) Cytoplasmic and nuclear fractions from efferocytic and non-efferocytic BM macrophages were analyzed by Western blotting using HIF-1α, p-STAT3, and Histone H3 antibodies. The graph shows independent experiments where nuclear HIF-1α and p-STAT3 intensities were normalized to total protein and to the corresponding non-efferocytic BM macrophages (*n* = 2). Increased nuclear protein in the efferocytic fractions is shown for each experiment. (**B**,**C**,**H**,**I**) Representative Western blot images. Plotted data are shown as mean ± SEM, * *p* < 0.05, ** *p* < 0.01, *** *p* < 0.001, **** *p* < 0.0001, ns = not significant (ordinary one-way ANOVA; Tukey’s multiple comparison test and unpaired *t*-test). Additional results are shown in Appendix B—Figure A1, Figure A2, Figure A3 and Figure A4.

**Figure 3 cells-11-03712-f003:**
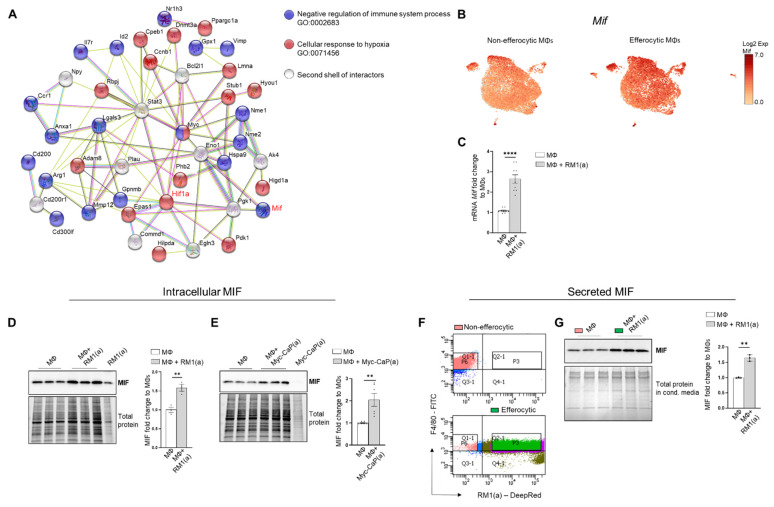
BM macrophage efferocytosis induces MIF expression. (**A**) Protein–protein interaction network between the immune pathway regulation and the hypoxia-related pathway. Biological Process GO terms of selected genes upregulated by efferocytosis (GO:0002683, all genes; GO:0071456, 25 out of 33 genes). BM macrophages (MΦ) were isolated from 4 wk old C57BL/6J or FVB/NJ mice and co-cultured with apoptotic RM1(a) or Myc-CaP(a) cells for 16–18 h. (**B**) scRNA-Seq analysis plot shows *Mif* distribution in efferocytic and non-efferocytic clusters of BM macrophages. (**C**) *Mif* mRNA expression in efferocytic BM macrophages assessed by RT-qPCR (*n* = 9). Western blot analysis of protein lysates from (**D**) C57BL/6J BM macrophages co-cultured with apoptotic RM1 prostate cancer cells (*n* = 3) and (**E**) FVB/NJ efferocytic and control BM macrophages were analyzed by Western blotting using MIF antibody (*n* = 6). (**F**) C57BL/6J efferocytic (F4/80^+^DeepRed^+^ (in green)) and control non-efferocytic BM macrophages (F4/80^+^ (in salmon)) were sorted, seeded, and incubated for 6 h. (**G**) Western blot analysis of conditioned media from non-efferocytic and efferocytic BM macrophages (*n* = 3/group). The Western blot shown in E is a representative image. Plotted data are shown as mean ± SEM of quantified signal after normalization to total protein and to control BM macrophages, ** *p* < 0.01, **** *p* < 0.0001 (unpaired *t*-test). Additional results are shown in Appendix B—Figure A5.

**Figure 4 cells-11-03712-f004:**
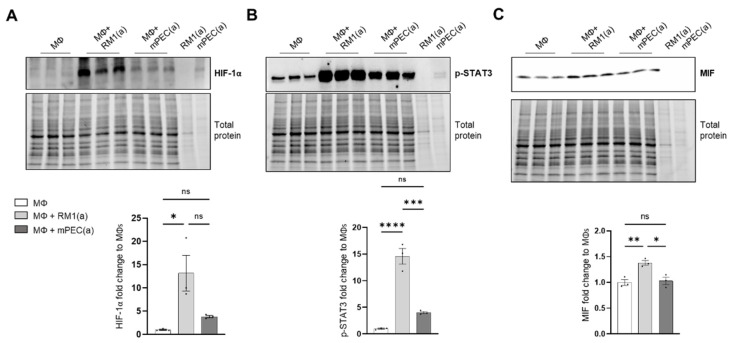
HIF-1α, p-STAT3, and MIF expression is increased in apoptotic RM1-efferocytic BM macrophages. BM macrophages (MΦ) were isolated from 4 wk old C57BL/6J mice and co-cultured with apoptotic RM1(a) or mPEC(a) cells for 16–18 h. (**A**–**C**) Western blot analysis of protein lysates using HIF-1α, p-STAT3 and MIF antibody (*n* = 3). Plotted data are shown as mean ± SEM, * *p* < 0.05, ** *p* < 0.01, *** *p* < 0.001, **** *p* < 0.0001, ns = not significant (ordinary one-way ANOVA test).

**Figure 5 cells-11-03712-f005:**
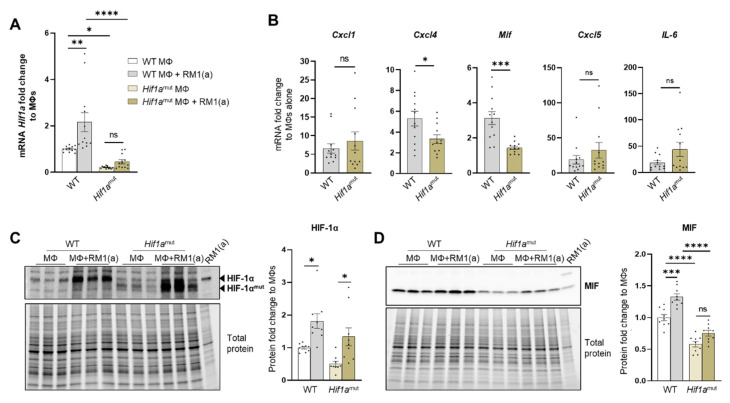
HIF-1α depletion in efferocytic BM macrophages reduces MIF expression. (**A**) *Hif1a* mRNA expression levels in *Hif1a*^flox/flox^ (WT) and *Hif1a*^mut^ BM macrophages (MΦ). (**B**) mRNA from efferocytic and control BM macrophages from *Hif1a*^flox/flox^ and *Hif1a*^mut^ mice were analyzed by RT-qPCR for the specified inflammatory cytokine genes (*n* = 12). (**C**,**D**) Protein lysates from efferocytic and control BM macrophages from *Hif1a*^flox/flox^ (WT) and *Hif1a*^mut^ mice were analyzed by Western blot with HIF-1α and MIF antibodies (*n* = 9). The Western blots shown in (**C**,**D**) are representative images. Fold change was calculated relative to WT non-efferocytic BM macrophages (**A**,**C**,**D**) or to its own non-efferocytic BM macrophage control (**B**). Plotted data are shown as mean ± SEM. * *p* < 0.05, ** *p* < 0.01, *** *p* < 0.001, **** *p* < 0.0001, ns = not significant (ordinary one-way ANOVA; Tukey’s multiple comparison test and unpaired *t*-test). Additional results are shown in Appendix B—Figure A6, Figure A7 and Figure A8.

**Figure 6 cells-11-03712-f006:**
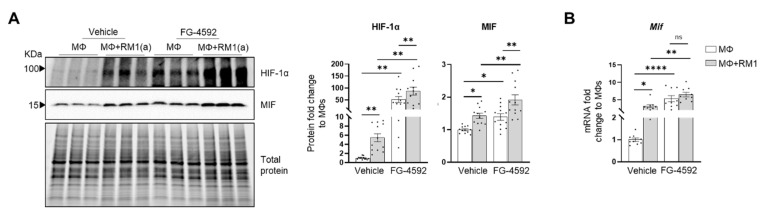
Efferocytic BM macrophages stabilize HIF-1α and induce MIF expression. BM macrophages (MΦ) were isolated from C57BL/6J mice and co-cultured alone or with apoptotic RM1(a) cells and treated with FG-4592 (Roxadustat, a HIF prolyl-hydroxylase inhibitor (10 µM)) or vehicle control for 16–18 h. (**A**) Protein lysates from co-cultures were analyzed by Western blotting with HIF-1α and MIF antibodies (*n* = 12/group). (**B**) mRNAs were isolated from co-cultures and analyzed by RT-qPCR for *Mif* expression (*n* = 9/group). Graphs show the fold change of each group relative to vehicle-treated BM macrophage control. The Western blots shown in A are representative images. Plotted data are shown as mean ± SEM; * *p* < 0.05, ** *p* < 0.01, **** *p* < 0.0001, ns = not significant (repeated measures one-way ANOVA; Tukey’s multiple comparison test).

**Figure 7 cells-11-03712-f007:**
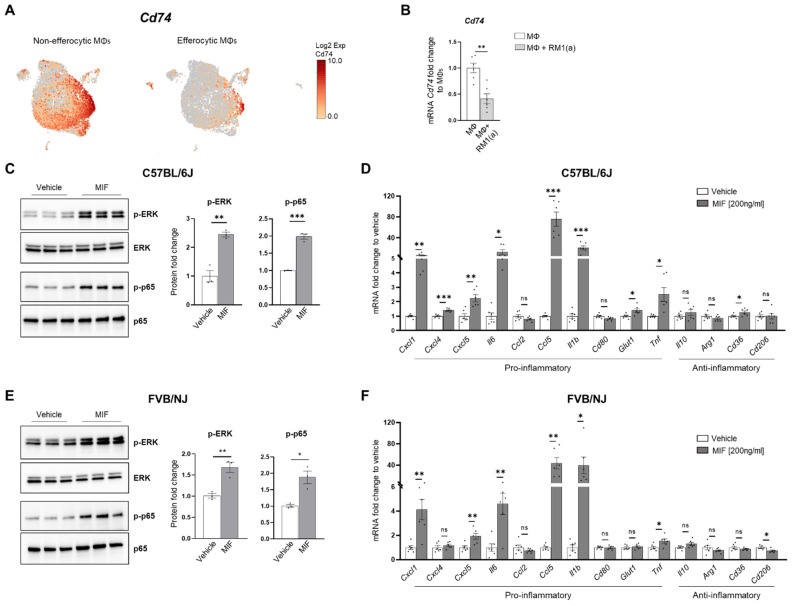
MIF induces a pro-inflammatory response in BM macrophages. BM macrophages (MΦ) were isolated from C57BL/6J mice and treated for 2 h with 200 ng/ml of MIF protein or vehicle control for Western blot analysis and 8 h for mRNA analysis. (**A**) scRNA-Seq analysis plot shows *Cd74* distribution in efferocytic and non-efferocytic clusters of BM macrophages. (**B**) *Cd74* mRNA expression in whole efferocytic BM macrophages assessed by RT-qPCR (*n* = 6/group). Protein lysates from BM macrophages (**C**) C57BL/6J and (**E**) FVB/NJ treated with MIF (200 ng/ml) and vehicle control were analyzed by Western blotting with total ERK, p-ERK, total p65, and p-p65 antibodies (*n* = 3/group). mRNAs were isolated from BM macrophages (**D**) C57BL/6J and (**F**) FVB/NJ treated with MIF and vehicle control and analyzed by RT-qPCR for the specified pro- and anti-inflammatory factors (*n* = 6/group). Data are shown as mean ± SEM; * *p* < 0.05, ** *p* < 0.01, *** *p* < 0.001, ns = not significant (unpaired *t*-test). Additional results are shown in Appendix B—Figure A11.

## Data Availability

Data sets of raw sequencing data for both experiments: Experiment 1 (GSM5466517/roca4, non-efferocytic and GSM5466518/roca5, efferocytic) and Experiment 2 (GSM5466519/roca6, non-efferocytic and GSM5466520/roca7, efferocytic) are deposited in the NCBI Sequence Read Archive (SRA) and can be accessed from the NCBI Gene Expression Omnibus (GEO, Series Accession: GSE180638).

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
