# Peer review of "Bone Marrow Macrophages Induce Inflammation by Efferocytosis of Apoptotic Prostate Cancer Cells via HIF-1α Stabilization"

_cells, 2022, doi:10.3390/cells11233712_

Round 1

Reviewer 1 Report

The authors of this publication describe that efferocytosis of apoptotic prostate cancer cells stabilizes HIF1α in macrophages. This in turn stabilized MIF, which might then induce paracrine effects on macrophages that have not been involved in the efferocytic process.

The finding that efferocytosis can induce HIF1α in macrophages is new, although a very recent publication already linked this process to HIF1α (DOI: 10.1080/17435390.2022.2083995). Nonetheless, the manuscript in its current state fails to persuade me that the herein described processes really induce “chronic inflammation in the metastatic bone microenvironment” (l. 625) as the authors claim in their discussion. To obtain this goal, there should experiments be added that analyze the effect of conditioned medium of efferocytic macrophages (with and without functional HIF-1α) on “effector” cells (which might be non-efferocytic macrophages or other immune cells). As a first step, the authors must show an increased MIF release of the efferocytic cells (please see major point 6).

Major points:

1.       Three days of macrophage differentiation are quite a short time period. The authors should provide a flow cytometry analysis of F4/80 in their macrophage culture to ensure the purity of the macrophage culture.

2.       Figure 2A: Why did efferocytosis stabilize Hif1a mRNA under normoxic conditions? Did efferocytosis induce the NF-kB pathway? There is a well-known interaction of p65 with the Hif1a promoter. The authors should add data and discussion on this point.

3.       It is well-known that STAT3 can stabilize HIF-1α. The authors show in Figure 2 D-G that this pathway plays a role in macrophages as well. Nonetheless, the blocking of STAT3 did not completely abrogate the stabilization of HIF-1α. The authors should comment on possible reasons for that.

4.       The HIF-1α-ubiquitin CoIP is not persuasive to me. How can a densitometric analysis be made from such a smeary and unconcrete signal? With reference to my point 3 above I would delete Figure 2H. In addition, it shows data with n=1. This is not adequate for publication (the same is true for Figure 2 I – if these are representative blots for replicated experiments this should be indicated in the figure legend).

5.       Ll. 401-405: The text does not fit to the description in the figure. Please clarify which is the HIF-1α blot and which one is the pSTAT3 blot. Furthermore, in figure 4B the standard deviation given in the graph below does not fit to the WB above, this is obviously higher for the blot shown in 4B as in 4A.

6.       Figures 5 and 6: The authors should analyze whether the induction in MIF protein also leads to an induction in MIF release. Otherwise they cannot proclaim paracrine effects later on.

7.       Figure 6: Can the authors speculate on why Roxadustat treatment can induce MIF protein expression in efferocytic macrophages but fails to induce the mRNA?

8.       Figure 7: What is the effect of MIF treatment on the expression of Hif1a/HIF-1α??? As p65 is induced and Hif1a is a direct target gene of classical NF-kB activation one would expect Hif1a to be upregulated as well. In addition: Why did the authors analyze protein and mRNA expression with such a big difference in time (6 hours according to ll. 126/127)?

9.       Figure 7A: What is the mechanism and the function of the downregulation of CD74 in efferocytic cells?

10.   L. 625: Where did the authors analyze CHRONIC inflammation in this manuscript? This sentence must be revised as it is overinterpreting the given data.

11.   Figure 5.2: Why do the Epas1 k.o. mice show (highly significantly!) reduced levels of Hif1a mRNA? And why does this not correspond to changes in the protein expression of HIF-1α? Could the authors comment on that? In addition, the k.o. efficiency of Epas1 should be given in this figure or at least in the M&M section of the manuscript.

Minor points

1.       The authors use the terms: bone macrophages - bone marrow macrophages - bone marrow derived macrophages synergistically throughout the manuscript; please unify  

2.       The loaded amounts of protein should be given in the figures showing Western Blot data.

3.       Neither HIF-1α nor HIF-2α are transcription factors. Transcriptional activity requires the dimerization with HIF-1β and the transcription factors are then termed “HIF-1” and “HIF-2”.

4.       Why did the authors use different concentrations of MIF in figure 7 and figure 7.1?

Reviewer 2 Report

The authors of the Mendonza-Reinoso work did very detailed work on the signaling pathway involving the HIF/NFKB/MIF axis. The article is well written and the experimental approach is well outlined. In general, the literature on apoptosis and efferocytosis has shown for decades that these elements are strongly associated with anti-inflammatory/regulatory/M2 processes (Pubmed 30291029, 31822793, 32251387). Furthermore, the role of macrophages (TAM) in the maintenance of the tumor process is well known, they present a polarization between M1/M2. Moreover, HIF-α/MIF and NF-κB/IL-6 axes have been previously shown as important processes to recruit leukocytes to cancer, favoring the growth of the tumor process in head and neck squamous cell carcinoma (24709424). In this context, it seems contradictory to me for the authors to report that they observed a pro-inflammatory response and at the same time it was immunosuppressive. My main suggestions from improve this work:

1) The authors mainly evaluated the expression of chemokine, and IL-6, which can be both pro and anti-inflammatory. I recommend that a deeper analysis of the inflammatory response of macrophages be carried out. The authors should evaluate the expression of classic markers of macrophage polarization, both pro-inflammatory (IL1, INOS/NOX2, CD80, glut1) and anti-inflammatories (IL10, Arg-1, cd206, and cd36), whether by PCR or Wb. As far as possible, I recommend analyzing at least two markers for each type of response.

2) What processes would be behind the induction of a different response between apoptotic bodies of tumor cells about normal cells? I strongly suggest that the authors evaluate oxidative stress in this process, which is common in tumor processes.

3) MIF has a very strong immunosuppressive character in lymphocytes (PMID 24709424, PMID: 29864117) and pro-angiogenic (PMID: 33542244). I suggest that these topics be addressed in the discussion

Round 2

Reviewer 1 Report

The authors have addressed my concerns and have indeed increased the quality of the manuscript to a great extent.

Reviewer 2 Report

The article addresses all my doubts and suggestions, I believe it is a good acquisition for this journal.